# Progress on New Preparation Methods, Microstructures, and Protective Properties of High-Entropy Alloy Coatings

Kefeng Lu [1], Jian Zhu [1,2,3,*], Wenqing Ge [1,*] and Xidong Hui [4]

1 School of Mechanical Engineering, Shandong University of Technology, Zibo 255000, China
2 Shandong Highland Oil & Gas Equipment Co., Ltd., Dongying 257091, China
3 Shandong Provincial Key Laboratory of Precision Manufacturing and Non-Traditional Machining, Zibo 255000, China
4 State Key Laboratory for Advanced Metals and Materials, University of Science and Technology Beijing, Beijing 100083, China
* Correspondence: zhujian@sdut.edu.cn (J.Z.); gwq@sdut.edu.cn (W.G.)

**Abstract:** Currently, the preparations of high-entropy alloy (HEA) coatings have developed into new methods such as thermal spraying, electrospark deposition technology, and magnetron sputtering. The microstructures and protective properties of HEA coatings prepared by different methods are bound to be different. Moreover, because HEAs have a wide range of composition systems, the difference in composition will inevitably lead to a change in process parameters and post-treatment methods, and then affect the microstructures and protective properties. This paper introduces the working mechanism of thermal spraying, electrospark deposition technology, and magnetron sputtering, compares the advantages and disadvantages of each method, and focuses on the influences of the compositions, process parameters, and post-treatment process on the microstructures and properties of the coating. Furthermore, this paper outlines the correlation between preparation methods, process parameters, microstructures, and properties, which will provide a reference for further development of the application of high-entropy alloy coatings. On this basis, the future development direction of HEA coatings is prospected.

**Keywords:** high-entropy alloys; coating; thermal spraying technology; electrospark deposition technology; magnetron sputtering technology

## 1. Introduction

Multi-principal HEAs break with the traditional design principle of alloy materials with one metal element as the primary group element [1]. Among them, entropy is a physical quantity representing the disorder degree of the system. High-entropy materials include HEAs and high-entropy ceramics [2]. With the increase in mixing entropy, the change in the Gibbs free energy decreases, which leads to the phase stability of solid solutions [3]. Although HEAs are composed of multiple group elements, they form simple FCC, BCC, or HCP solid solutions after solidification [4–7]. Compared to conventional alloys, HEAs have some outstanding features by virtue of their innovative and unique design concepts [8]. The four major effects of HEAs allow for superior properties such as wear and corrosion resistance as well as oxidation resistance [9–20]. Surface coatings are used to protect and enhance the life of structural components in adverse operating conditions [21–23]. The application of HEAs as coating materials is a major direction for the development of HEAs. Intermetallic compounds are easily produced during the preparation of HEAs due to the slow cooling rate, which affects the alloy properties, while the coating is capable of rapid cooling due to its low thickness, thus facilitating the acquisition of higher-quality alloys [24–30]. Currently, the most commonly used method to prepare HEA coatings is the laser melting technique, but the heat input of laser melting is high and prone to defects such as microcracks and porosity. Therefore, thermal spraying,

EDM (EDM, electrical discharge machining) deposition, and magnetron sputtering, an emerging technology technique to prepare HEA coatings, have come into being.

Thus far, there are fewer summaries and generalizations about the preparation of high-entropy alloy coatings and coating organization and properties by new methods such as thermal spraying, EDM deposition, and magnetron sputtering. Moreover, most of them concentrate on the discussion of optimal process parameters, lacking a more comprehensive summary of the influence mechanism of each process parameter. Meanwhile, there is a lack of generalization on the effect of HEA composition modulation and post-treatment methods on coating properties. As a result, this paper centers on the progress of research on the preparation of HEA coatings by new methods such as thermal spraying, EDM deposition, and magnetron sputtering, introducing the working mechanism, and advantages and disadvantages of the three processes. This article highlights the effects of three processes, HEA coating composition regulation, process parameters, and post-treatment measures, on the coating organization and properties. On this foundation, the development of the HEA coating is summarized and prospected.

## 2. Novel Preparation Method of HEA Coatings

There are two main methods for preparing HEA coatings: hot forming and cold forming. Among them, the main processes of hot forming include laser cladding and thermal spraying. The working principles of these two methods are similar. This method has the advantages of simpler operation, wider preparation area, and on-site preparation. However, the requirements of the powder particle size and flowability are more stringent, and particles are prone to oxidation in a high-temperature environment. The main processes of cold forming are EDM deposition and magnetron sputtering technology. Electric spark deposition technology uses the substrate material as the cathode and the electrode as the anode, which generates electric sparks when the substrate and the electrode are in contact to release a large amount of energy, causing the electrode material to melt, vaporize, or turn into a plasma. Magnetron sputtering generally involves bombarding the substrate with ions under the action of an electric field, which blasts ions, atoms, and molecules from the surface of the substrate and then shoots them at a certain speed to form a coating. The coatings prepared by this type of process have a homogeneous microstructure and excellent properties; however, they require more demanding working conditions, are not easily prepared on-site, and have a limited thickness.

## 3. Thermal Spraying Technology

As the technology of preparing HEA powder by the mechanical alloying method, powder metallurgy method, and aerosolization matures, the thermal spraying methods for preparing HEA coatings will become more popular, among which the most commonly used methods are flame spraying, plasma spraying, and electric arc spraying. Among them, the heat source temperature of plasma spraying can reach 10,000 °C or more, spraying a wide range of materials, so it is the most widely used thermal spraying technology. Figure 1 shows the working principle of thermal spraying. The coating is formed by mechanical impact of the coating material to the substrate. The specific process is to first use a heat source (such as plasma, flame, and arc) to heat the coating material to a molten or semi-molten state, and the coating material is then sprayed onto the surface of the substrate by means of a high-pressure gas at a certain speed. Many studies have shown that thermal spraying enables effective preparation of HEA coatings. The performance of HEA coatings varies considerably by composition. The process parameters of thermal spraying also have an important effect on the forming quality of the coating. In addition, heat treatment can also effectively improve the hardness, corrosion resistance, and other properties of the coating.

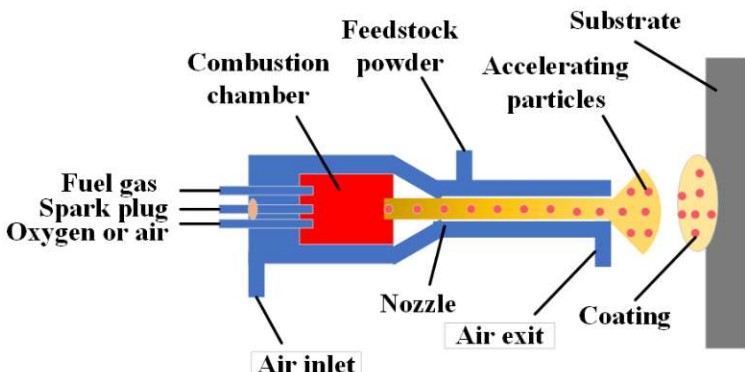

**Figure 1.** The basic mechanism of the thermal spray process.

### 3.1. System of the HEA Coating

Plasma spraying technology has the advantages of high energy density, low thermal distortion, low dilution rate, etc., which has attracted more attention in the preparation of HEA coatings. At present, researchers have carried out a lot of research on the different systems of HEA coatings formed by plasma spraying. The TiAlNb intermetallic compound with high temperature and low density has been successfully synthesized by using the element of Nb, and the large atomic size is conducive to the formation of the second phase. Liu et al. [31], by studying the effect of Nb on the CoCrFeNi HEA coatings prepared by plasma spraying, found that the addition of Nb elements transformed the alloy structure from FCC to HCP, and the yield and fracture strengths gradually increased. Cheng et al. [32] studied the effect of Nb elements on CoCrCuFeNi HEA coatings based on plasma spraying, and the addition of Nb elements can effectively improve the wear resistance of the alloy. Wang et al. [33] studied the high-temperature friction and wear properties of (CoCrFeMnNi)85Ti15 HEA coatings prepared by plasma spraying, and found that the coating microstructure consisted of FCC and BCC. In addition, the addition of Ti elements effectively improved the high-temperature wear resistance of the alloy, as seen in Figure 2, with the best wear resistance at 400 °C.

Supersonic flame spraying uses high-pressure oxygen and fuel in a combustion chamber to create a flame stream that melts and sprays HEA powder onto the surface of the substrate to form the coating. Because of the fast particle spraying speed, the coating organization is denser than plasma spraying, and fewer oxides are produced during the spraying process. Chen et al. [34] investigated the wear resistance of supersonic-flame-sprayed Al0.6TiCrFeCoNi HEA coatings at different temperatures and the study showed that the coating had a single BCC phase and that the increased temperature reduced the friction coefficient of the coating. Lobel et al. [35] investigated the wear behavior of AlCoCrFeNiTi0.5 HEA coatings prepared by supersonic flame spraying at high temperature. The coating was found to form a body-centered cubic phase with B2 and A2 structures. With the increase in temperature, an oxidation protective film was formed, and the wear resistance was also enhanced. Adding ceramic particles to the coating can effectively enhance the hardness, wear-resistance, and corrosion resistance of the coating. Vallimanalan et al. [36] investigated the corrosion behavior of supersonic flame spraying Mo elements on AlCoCrNi HEA coatings and compared the corrosion performance with conventional NiCrSiB-corrosion-resistant coatings. The results showed that the AlCoCrMoNi HEA system had better corrosion resistance than NiCrSiB coatings. Tang et al. [37] studied the effect of ceramic particle addition on the corrosion resistance of supersonic-flame-sprayed stainless-steel coatings, and the results showed that the coarse-grained $Al_2O_3$ composite coating had a porosity of 0.7863% and a hardness of 637 HV0.1 with good corrosion resistance, and the fine-grained $Al_2O_3$ composite coating had a hardness of 600 HV0.1, which also had good corrosion resistance.

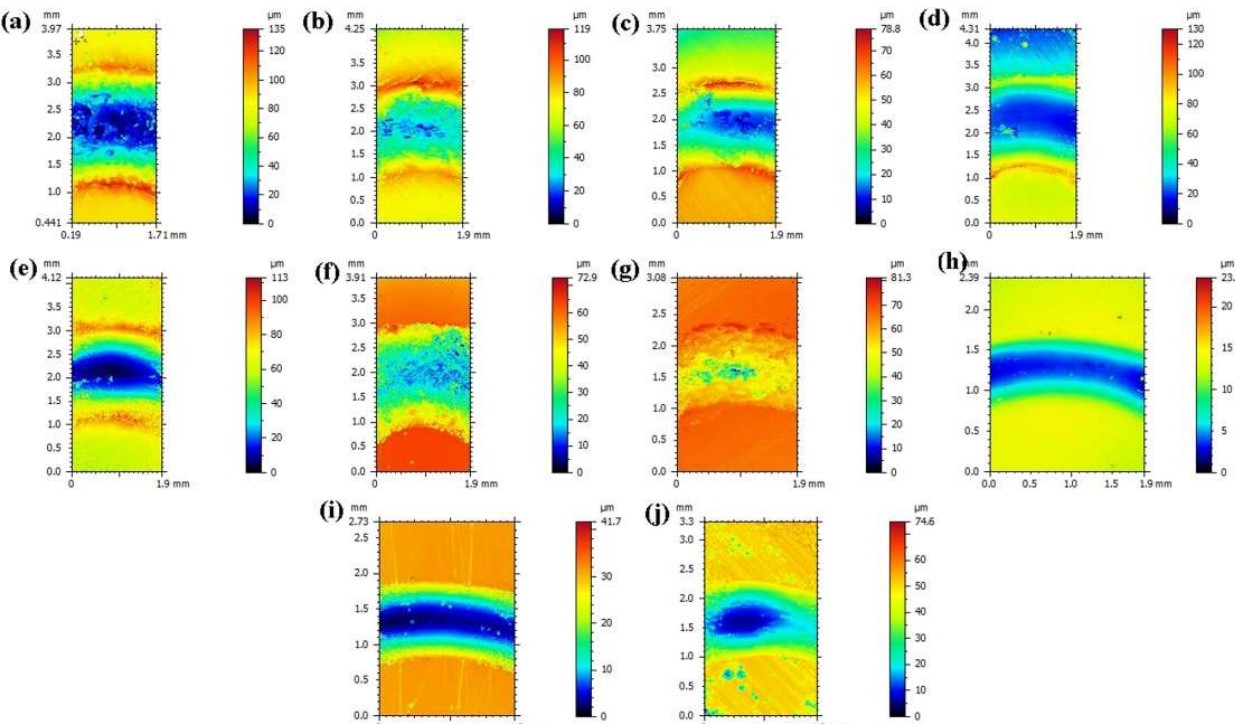

**Figure 2.** Wear surface profilometry of CoCrFeMnNi (**a–e**) and (CoCrFeMnNi)85Ti15 (**f–g**) coating at: (**a,f**) room temperature, (**b,g**) 200 °C, (**c,h**) 400 °C, (**d,i**) 600 °C, and (**e,j**) 800 °C. (Reprinted with permission from ref. [33]. Copyright 2020 Elsevier).

High-speed arc spraying uses a short arc to melt the metal wire and then uses a high-pressure gas jet on the surface of the matrix to form a coating. Due to the fact that particles are easy to oxidize, the spraying speed is low, and the phase transformation of the coating is difficult to control, there are few studies on the preparation of HEA coatings by high-speed arc spraying. However, this method is suitable for the preparation of large industrial coatings and has high industrial development potential. Guo et al. [38] prepared FeCrNiCoCu and FeCrNiCoCuB HEA coatings on the surface of AZ91 magnesium alloy by high-speed arc spraying technology. The results showed that the two HEA coatings formed densely layered structures, and the microhardness of the two coatings reached 408 and 346 HV, respectively.

Adding some metallic or nonmetallic elements has a certain effect on the coating. The HEA coatings exhibit excellent and unique properties due to the thermal spray preparation of different element types. The applications of HEA coatings prepared by thermal spraying are mainly focused on corrosion-resistant coatings, wear-resistant coatings, and high-temperature-oxidation-resistant coatings. The development of new HEA spraying materials can be of great help to broaden the HEA coatings.

*3.2. The Process Parameters of Thermal Spray*

Due to the complex arrangement of HEA atoms, its microstructure and properties are more easily affected by the preparation process, so it is of great significance to optimize the process parameters of thermal spray. Wei et al. [39] studied the influence of plasma cladding process parameters on the microstructure and properties of CoCrFeMnNi HEA coatings. The results showed that the influencing factors on the dilution rate of the cladding layer from small to large were the powder gas flow rate, arc current, and traveling speed, and the influences on the hardness of the cladding layer from small to large were powder gas flow rate, traveling speed, and arc current. Particles with high speed will form a dense coating. If the particle temperature is too low, the bonding strength between the coating and substrate could be insufficient. Particles in the half-molten state could improve the bonding

strength. Therefore, studying the particle behavior is helpful to obtain the optimal process parameters, leading to good performance of the coating. Ang et al. [40] used DPV-2000 to detect the flight speed, temperature, and particle size in plasma spraying. The results showed that the melting state and flight speed of particles could be controlled by reasonable process parameters, to improve the microstructure and properties of the coating. Wang et al. [41] studied the particle flight behavior in supersonic flame spraying by using fluid dynamics software Fluent, and the results showed that the optimal particle size range was 30–50 μm. The injection velocities for particles should be optimized for different particle sizes. The optimal injection velocities for particles with small, medium, and large particle sizes were 10–15 m·s$^{-1}$, 5–10 m·s$^{-1}$, and 1–5 m·s$^{-1}$, respectively. Compared with spherical particles, nonspherical particles can acquire less heat and more kinetic energy during flight.

In plasma spraying, the cladding current directly affects the molten pool temperature; determines the melting state of HEA powder, the growth time of grains, and the diffusion rate of elements in the molten pool; has an important influence on the properties, bonding strength, and microstructure of the coating with the substrate. If the cladding current is too large, the matrix dilution rate will increase and the toughness will be reduced. On the contrary, the coating will have defects such as pores and inclusions. Zhang et al. [42], by studying the effect of cladding current on the coating, found that under the current of 90, 100, and 110 A, the coating was thin, the matrix dilution rate was low, and pores appeared when the current was small. With the increase in current, the coating microstructure gradually improved, but local accumulation occurred when the current was too large. Wei et al. [43] studied the effect of cladding current on the microstructure and properties of a FeCoCrNiMn HEA coating. It was found that the current had no effect on the FCC phase of the coating, but had a significant effect on the microstructure. With the increase in cladding current, the dendrite microstructure became coarser and the hardness of the coating decreased. Tian et al. [44] studied the influence of spraying current on the microstructure and properties of the high-speed arc spraying coating, and the results showed that spraying current had a great influence on the microstructure density and bonding strength of the coating. The FeNiCrAl HEA coating obtained by a spraying current of 200 A, spraying distance of 160 mm, and voltage of 34 V had a compact microstructure, low porosity, bonding strength of 52.3 MPa, hardness of 626 HV0.1, and about 1.6 times the matrix hardness. The FeNiCrAl HEA coating obtained by a spraying current of 200 A, spraying distance of 160 mm, and voltage of 34 V had a compact microstructure, low porosity, bonding strength of 52.3 MPa, hardness of 626 HV0.1, and about 1.6 times the matrix hardness.

The moving speed of the spray gun directly determines the accumulation of coating powder and affects the thickness and width of the coating. The gun movement speed should be adjusted according to the required coating thickness. Wang et al. [45] studied the in situ NbC/HEA microstructure of the plasma cladding and found that the coating performance was the best when the moving speed of the spray gun was 7 mm/s. Jinbin et al. [46] used a plasma cladding to prepare a CoCrCuFeMnNi HEA coating on a Q235 steel substrate under the process parameters of a 10 mm distance between the nozzle and workpiece surface and a scanning speed of 200 mm/min, and obtained a HEA coating with a thickness of 1 mm without cracks and pores.

In thermal spraying, particle velocity, temperature, and particle size have important effects on the coating dilution rate, density, and matrix bonding strength. The cladding current directly affects the molten pool temperature, the melting state of HEA powder, the growth time of grains, and the diffusion rate of elements in the molten pool. The moving speed of the spray gun directly determines the amount of coating powder accumulation and affects the thickness and width of the coating. In addition, the size of the powder is also one of the key factors affecting the forming quality of the coating; if the powder is too small, it will make the coating thickness and the height of the spray gun low, which is not conducive to the formation of the molten pool. On the contrary, it will make the

powder splash serious, causing a low material utilization rate and resulting in uneven coating organization.

### 3.3. Post-Processing

Thermal spraying is based on spraying a liquid metal alloy in the form of wire or powder to feed the plasma gun. It improves the corrosion, oxidation, and wear resistance of the substrate while also providing insulation. As a result, the life of the protected components is extended. However, the evolution of the coating-related microstructure after the thermal spraying process remains uncertain. Wang [47] investigated the thermal spraying NixCo0.6Fe0.2CrySizAlTi0.2 HEA coating organization and strengthening mechanism. The results showed that the hardness of the HEA coating after 1100 °C/10 h heat treatment increased significantly to the casting state. Comparing the NixCo0.6Fe0.2CrySizAlTi0.2 casting structures with adding or removing Mn, Si, and Ni elements, Figure 3 shows the TEM images. It is seen that there were a large number of nanoparticles (5–10 nm), atomic segregation, entanglement structure, and subgrain structure in the matrix. However, after heat treatment at 1000 °C/10 h, the main characteristics observed in TEM were a large number of nanosized precipitates and dislocations. Ye et al. [48] investigated the high-temperature precipitation behavior and performance of the plasma-sprayed CoCrFeMnNi HEA coating. Figure 4 shows the surface scan images of the coating by different heat treatment processes. It was found that after 4 h of heat treatment at 700 °C, the Cr-rich δ phase (66Cr-13Fe-9Co-7Ni-5Mn) appeared at the grain boundary, and after 4 h of heat treatment at 500 °C, the grain boundary Cr-rich BCC phase (76Cr-9Fe-6Mn-5Co-4Ni) appeared, the micro-hardness of the coating increased and then decreased after heat treatment, and the wear quality loss of the melt layer changed in the opposite direction, with the highest hardness and best wear resistance of the coating heat-treated at 700 °C and the best corrosion resistance of the coating heat-treated at 500 °C in 3.5% NaCl solution. Liang et al. [49] investigated the effect of heat treatment on the organization and properties of a high-speed arc-sprayed FeCrNiCoCu HEA coating, and found that the coating underwent a crystallization transformation of the amorphous structure near 500 °C and that the NiCr-Fe phase appeared when the temperature exceeded 500 °C. The hardness of the coating increased with the increase in temperature and the peak value was reached at 500 °C, and the wear resistance was also significantly improved compared with that before heat treatment.

Through a reasonable heat treatment process can eliminate the defects generated during the preparation of the HEA coating by thermal spraying, it can also play a role in refining the grain, eliminating segregation, reducing internal stress, making the alloy organization more uniform, and improving the coating performance.

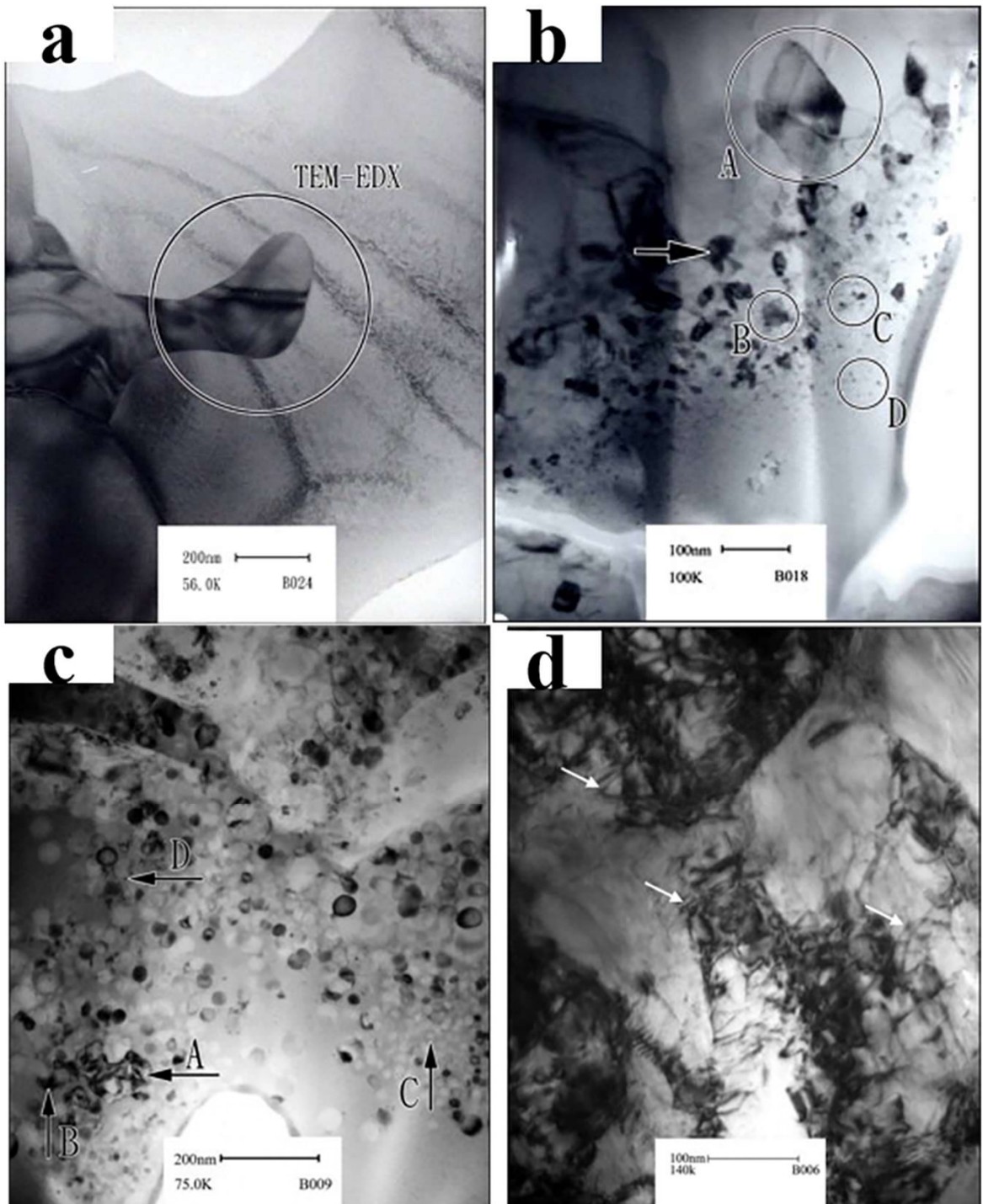

**Figure 3.** (**a**) TEM micrographs and EDS pattern of MnNiCo0.6Fe0.2Cr1.5SiAlTi0.2; (**b**) TEM micrographs of TS MnNiCo0.6Fe0.2Cr1.5SiAlTi0.2 following heat treatment at 1100 °C/10 h; (**c**,**d**) TEM micrographs of NiCo0.6Fe0.2CrSiAlTi0.2 at 1100 °C/10 h state. (Reprinted with permission from ref. [47]. Copyright 2011 Elsevier).

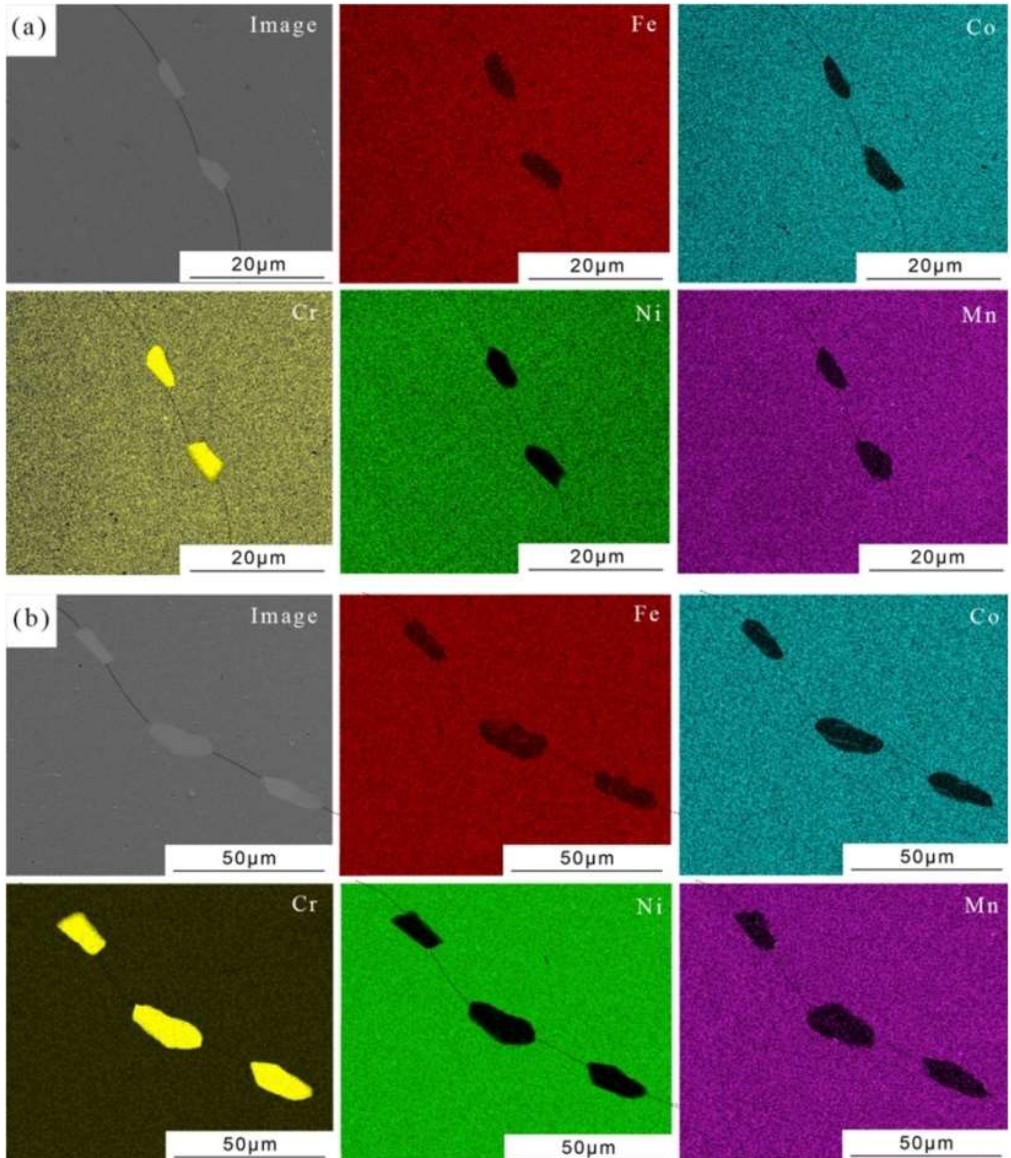

**Figure 4.** Surface scan image of the heat-treated cladding layer at (**a**) 500 °C; (**b**) 700 °C. (Reprinted with permission from ref. [48]. Copyright 2019 Elsevier).

## 4. Electrospark Deposition Technology

The electrospark deposition technology uses the matrix material as the cathode and the electrode as the anode. At the moment of contact between the electrode and the substrate, the high energy stored by the electrode produces sparks and releases a lot of energy. The contact position can produce a high temperature of 8000~25,000 °C to melt, gasify, or plasticize the electrode material, infiltrate the substrate surface, and form a coating on the substrate surface to achieve the purpose of surface strengthening or modification. Its working principle is shown in Figure 5. As an important surface modification technology, electrospark deposition technology has the advantages of rapid heating and rapid cooling, small heat input, metallurgical bonding between the deposited coating and substrate, a wide range of applications, and so on. In recent years, increasingly more studies have been conducted on the preparation of HEA coatings by EDM deposition technology.

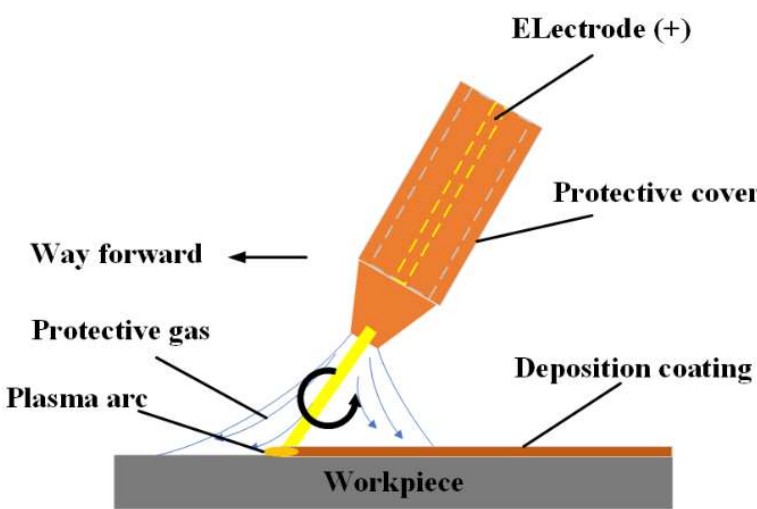

**Figure 5.** Schematic diagram for principles of electro-spark deposition.

*4.1. Composition System of HEA Coatings*

The only requirement for the electrode of the coating prepared by EDM deposition technology is electrical conductivity, so there is a wide range of optional materials. The research shows that a variety of different systems of HEA coatings can be prepared by EDM deposition technology, which can effectively improve the performance of the substrate. Chandrakant et al. [50] prepared AlCoCrFeNi HEA coatings by EDM deposition technology. It was found that the rapid solidification of small droplets led to the formation of a fine dendritic structure in the coating, and the uniformly diffused elements in the HEA formed an excellent coating–substrate interface. It can be seen from Figure 6 that the AlCoCrFeNi HEA coating effectively improved the hardness and wear resistance of the AISI410 stainless-steel substrate. Rongwang et al. [51] prepared CuNiTiZr HEAs on a TC11 titanium alloy substrate by the EDM deposition technique, and the XRD pattern of the coating is shown in Figure 7h. The microstructure of the coating changed with the number of deposited layers. The microstructure of the first- and third-layer coatings was composed of the BCC phase, and the microstructure of the coating changed from BCC to an amorphous state with the increase in the number of layers. The microstructure transformation was due to the rapid cooling rate and the increasing difference between the atomic size and the number of layers of the coating. All coatings were dense and homogeneous with no microcracks, as shown in Figure 7a–e. The content of individual elements varied gradually from the coating to the substrate (as shown in Figure 7f. This indicates that the bond between the coating and the substrate was a metallurgical bond. The average atomic percentage of each element in the coating is shown in Figure 7g. The average atomic percentage of each element varied with the number of layers. The large atomic radius difference between the main components of the coating facilitated the formation of disordered dense stacks, which can increase the solid–liquid interface energy, slow down the nucleation and growth of phases in the alloy, and lead to the amorphous formation. Li et al. [52] studied the microstructure and corrosion resistance of the AlCoCrFeNi HEA coating by EDM deposition technology. It was found that the coating structure had a simple BCC structure. Unlike the casting material, there was no inter-cellular segregation and nano-precipitated phase rich in Cr in the HEA coating, which had excellent corrosion resistance.

The electrospark deposition technology has the characteristics of rapid solidification. The preparation of HEA coatings by electrospark deposition technology can not only give full play to the high-mixing-entropy effect of multi-principal components, but also form a simple solid solution phase with a body-centered cubic structure or face-centered cubic structure, and the grain structure is fine, which plays the role of solid solution strengthening and fine grain strengthening, which is beneficial to obtain coatings with excellent properties.

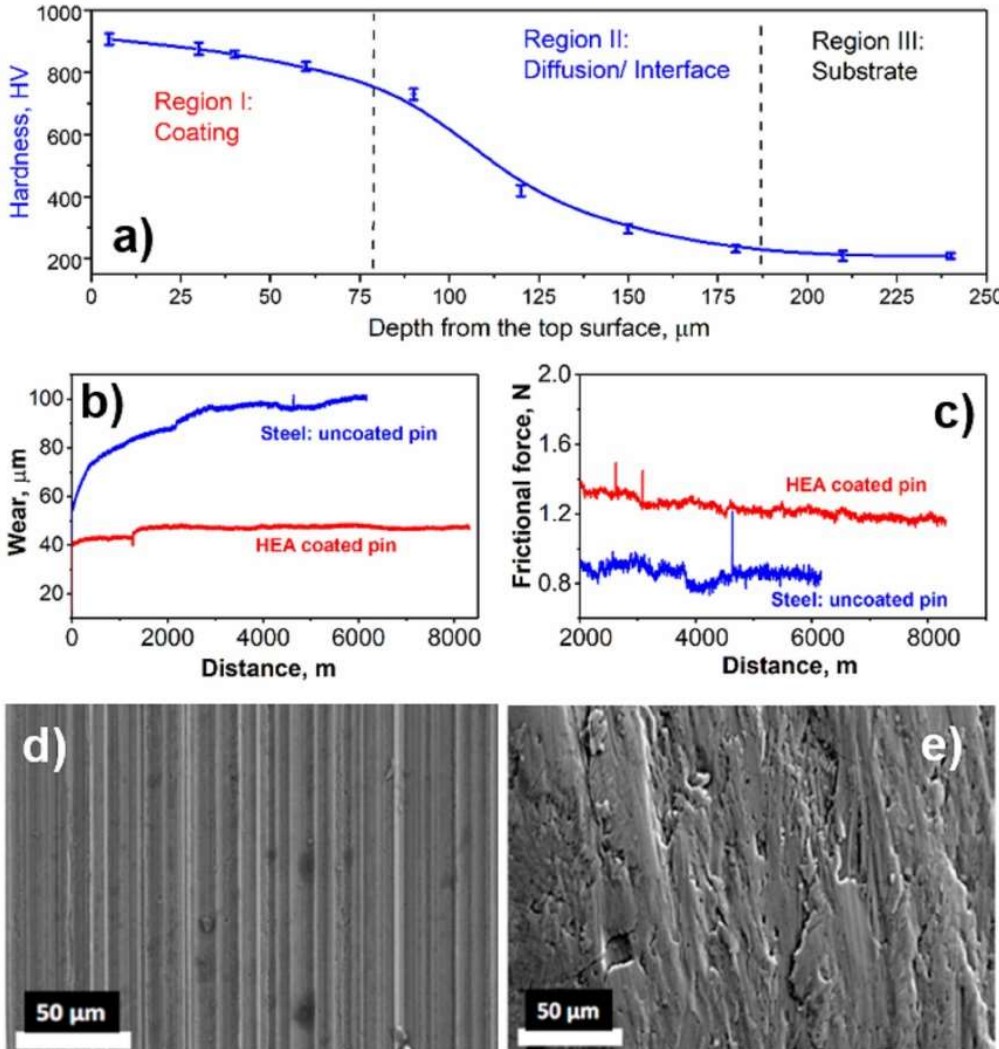

**Figure 6.** (**a**) Variation curve of hardness with depth at the cross-section of coating and substrate; (**b**,**c**) wet sliding wear behavior of pin specimens as a function of distance, before and after coating; (**d**,**e**) FESEM images of worn surfaces of substrate and coating samples after wear test. (Reprinted with permission from ref. [50]. Copyright 2021 Elsevier).

*4.2. The Process Parameters of Electrospark Deposition Technology*

The performance of HEA coatings can be improved by optimizing the technological parameters of electrospark deposition. Fuyu et al. [53] studied the effects of current density, plating solution temperature, and pH on the EDM deposition of the FeCoNiCr HEA coating. It was found that the coating was amorphous and uniformly coated, and the optimal process parameters were: current density 25 A·dm$^{-2}$, temperature 25 °C, and pH = 2.5. Yanfang et al. [54] studied the microstructure and corrosion resistance of the electrospark-deposited FeCoCrNiCu HEA coating. Through a large number of experiments, the optimum process parameters were obtained as follows: voltage 100 V, capacitive flow 10 μF, and frequency 150 Hz. The results showed that the prepared FeCoCrNiCu HEA coating can effectively improve the corrosion resistance of the substrate; Cean et al. [55] studied the high-speed friction and wear properties of an electrospark-deposited AlCoCrFeNi HEA coating with power 1000 W, argon flow rate 15 L/min, and specific deposition time 2.5 min/cm$^2$. The results showed that the coating had fine grains, compact structure, and no cracks, which was composed of BCC and FCC phases. The elastic modulus was reduced by about 8%, which had good wear resistance.

According to the discharge mechanism in the process of EDM deposition, the electric energy stored in the power supply is transformed into thermal energy in the form of a pulse, and the parameters such as deposition voltage, deposition capacitance, and strengthening time directly affect the properties of the coating. However, there are great differences in the electrical conductivity, thermal conductivity, and ductility of the material.

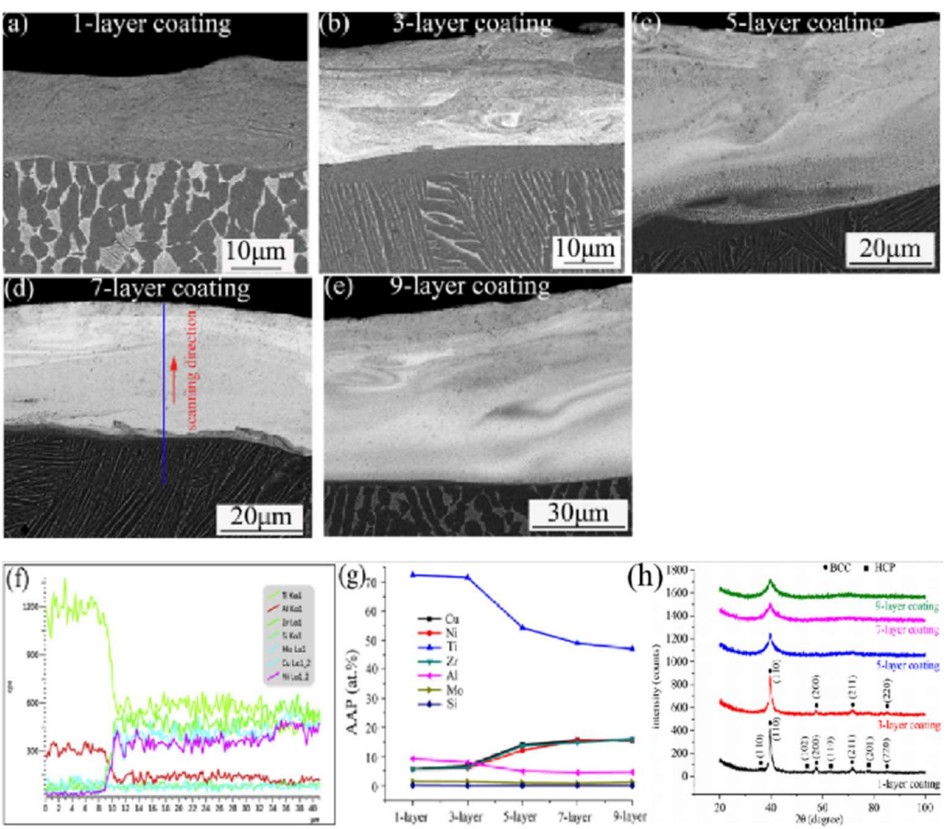

**Figure 7.** SEM(BSE) images of the CuNiTiZr MCACs: (**a–e**) 1/3/5/7/9-layer coating, respectively; (**f**) the line scanning result of the 7-layer coating; (**g**) EDS analysis.; (**h**) X-ray diffraction patterns of CuNiTiZr MCACs with various layers. (Reprinted with permission from ref. [51]. Copyright 2017 Elsevier).

### 4.3. Ultrasound-Assisted

To further enhance the performance of EDM-deposited HEA coatings, Yang et al. [56] proposed a new surface strengthening method—ultrasonic impact composite EDM surface strengthening technology. The ultrasonic impact causes severe plastic deformation and introduces a residual compressive stress field, improving the hardness, wear resistance, and corrosion resistance of the material. The principle of ultrasonic impact composite EDM surface strengthening technology is shown in Figure 8. The ultrasonic transducer converts the electrical signal from the ultrasonic generator into mechanical vibration of the same frequency, by applying downward static pressure on it, so the impact ball reciprocates between the surface of the workpiece and the output end of the ultrasonic transducer. Due to the voltage between the impact ball and the workpiece, there is a short-circuit contact between the impact ball and the workpiece. When the impact ball bounces from the surface of the workpiece, there is an electric spark between them, and the principle of the electric spark is the same as that of the welding arc. The coating obtained by ultrasonic impact composite EDM surface strengthening technology has excellent properties, but the research on the preparation of HEA coatings by this technology is very scarce.

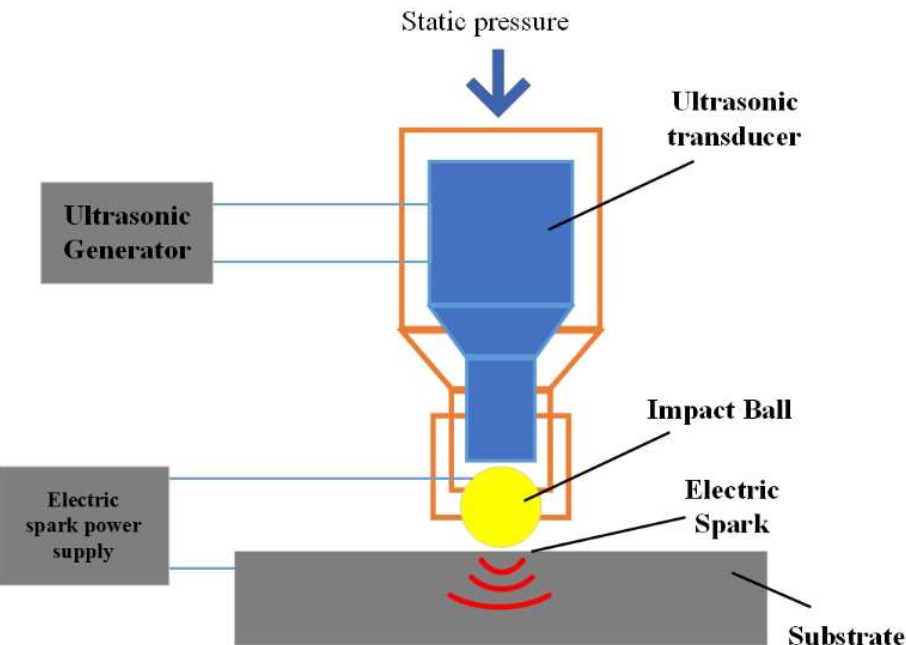

**Figure 8.** Principle diagram of ultrasonic impact composite EDM surface strengthening technology.

## 5. Magnetron Sputtering Technology

Magnetron sputtering generally involves bombarding the target with ions under the action of an electric field, bombarding the target surface with ions, atoms, and molecules, etc., and then shooting them at a certain speed at the substrate and depositing them on the surface to form a coating. Its working principle is shown in Figure 9. Magnetron sputtering has the advantages of high speed, low substrate temperature, and high adhesion between coating and substrate, which is one of the preferred processes in the preparation of HEA coatings.

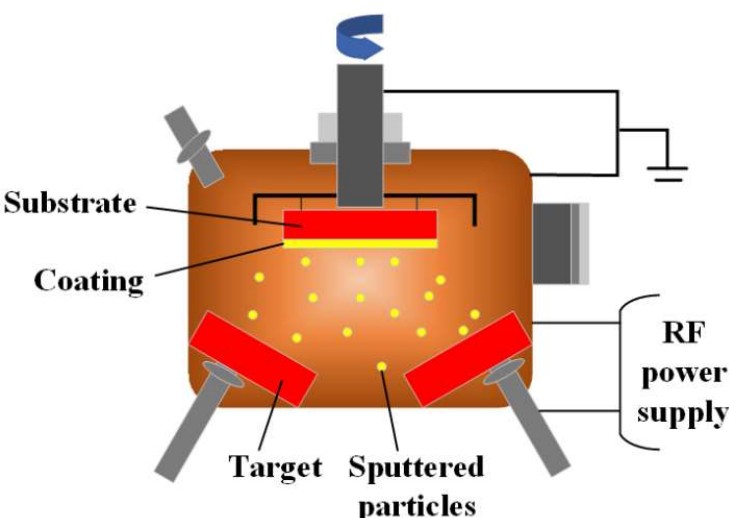

**Figure 9.** Schematic diagram of magnetron sputtering technology.

### 5.1. Composition System of HEA Coatings

At present, researchers have conducted a lot of research on the magnetron-sputtered HEA coatings of different systems. Liao et al. [57] prepared CoCrFeNiAl0.3 coatings by magnetron sputtering and studied their microstructure and properties. The results showed that magnetron sputtering was an effective way to prepare a high-quality nano-HEA coating with a smooth and uniform surface. As seen in Figure 10, the elastic modulus of

the coating was about 191 GPa, which is almost the same as that of a single crystal, and the nano-hardness was about 11.2 GPa, which is about 4 times that of the single crystal block. Zhao et al. [58] studied the mechanical and high-temperature corrosion properties of the AlTiCrNiTa HEA coatings prepared. It can be seen from Figure 11 that the coating structure was basically amorphous, with some FCC nanocrystalline areas. The coating had excellent high-temperature corrosion resistance. The addition of the element of Mo could promote the transformation from FCC phase to σ phase, thus improving the mechanical properties of the coating. Zhao et al. [59] added Mo elements to DC-magnetron-sputtered FeCoNiCrMn HEA, and the coating density increased and the phase structure gradually changed from a single FCC phase to a mixed phase of FCC + BCC phase. At the same time, the grain size decreased and the lattice constant increased. In our previous work, Mo-Ta-W RMPEA films, with a single BCC structure, were successfully prepared using the DC multi-target magnetron co-sputtering technique, and it was shown that the coating had a single BCC structure. The surface morphology of the films was lamellar and the cross-sectional structure was columnar dendrites. The coating had excellent mechanical properties, a hardness of ~20 GPa, and an elastic modulus of beyond 270 GPa [60].

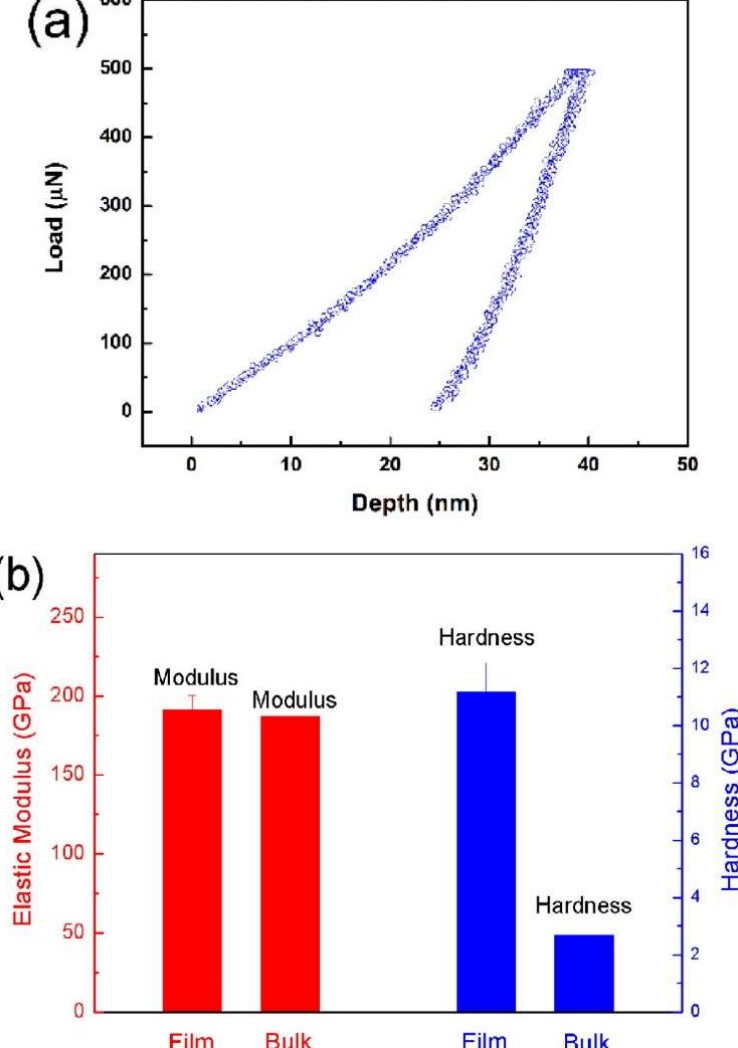

**Figure 10.** (**a**) Load–displacement curve of CoCrFeNiAl0.3 coating film test. (**b**) Hardness and elastic modulus of coatings and bulk samples. (Reprinted with permission from ref. [55]. Copyright 2017 Elsevier.)

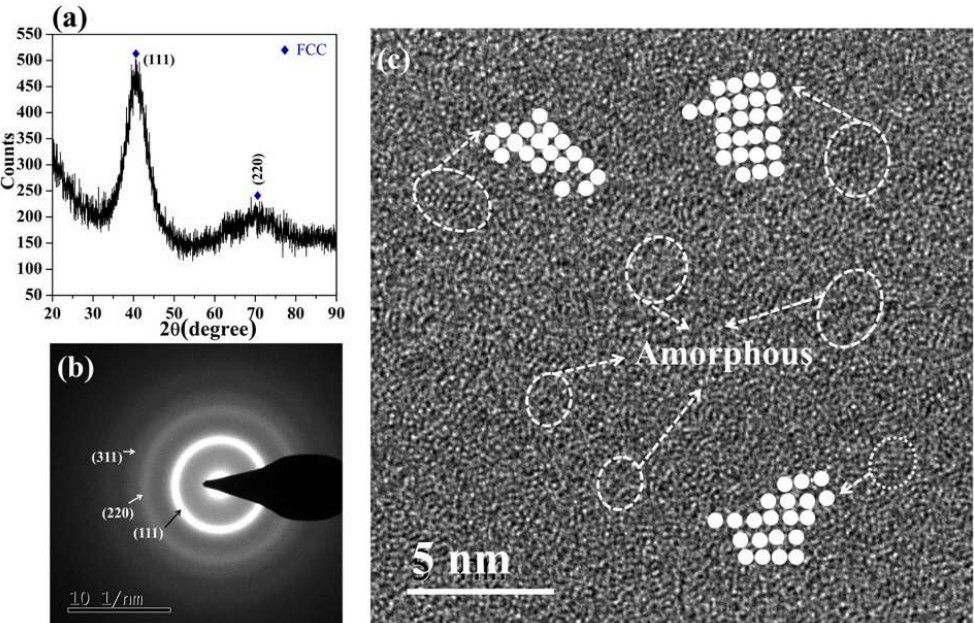

**Figure 11.** (**a**) GIXRD diagram of AlTiCrNiTa HEA coating; (**b**) SAED diagram of AlTiCrNiTa coating; (**c**) cross-sectional HRTEM diagram of AlTiCrNiTa coating. (Reprinted with permission from ref. [58]. Copyright 2021 Elsevier.).

### 5.2. The Process Parameters of Magnetron Sputtering Technology

The results showed that the microstructure and mechanical properties of the coating were significantly affected by adjusting the process parameters such as sputtering atmosphere, bias voltage, sputtering power, sputtering time, and substrate temperature. Chen et al. [61] studied the effect of $N_2$:Ar flux ratio on magnetron-sputtered AlCrTiZrVNx HEA coatings; when sputtered at a low $N_2$:Ar flux ratio, the AlCrTiZrVNx HEA coating showed an amorphous state, and hardness and elastic modulus were at low values. As shown in Figure 12, with the increase in $N_2$:Ar flux ratio, the XRD spectra showed (200) and (220) peaks in the XRD spectrum, and the (200) crystalline surface grew preferentially, indicating that the crystallinity of the film increased. When the $N_2$:Ar flux ratio was 3:4, the hardness and elastic modulus reached a maximum of 34.9 GPa and 323.8 GPa, respectively.

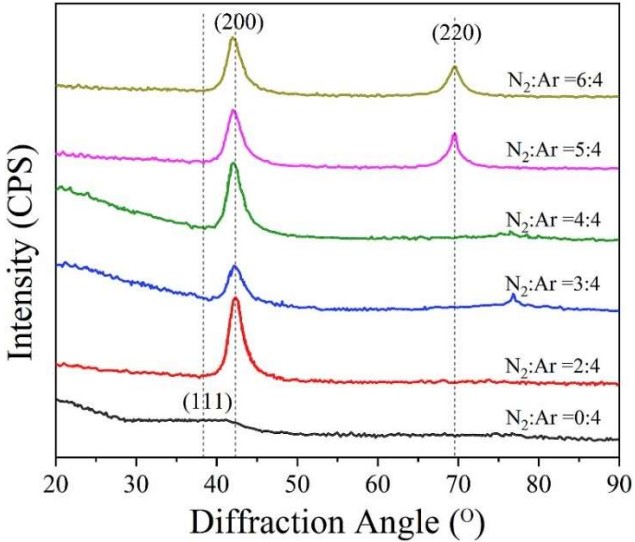

**Figure 12.** XRD diffraction pattern of AlCrTiZrVN*x* HEA coating under different $N_2$:Ar flow ratios. (Reprinted with permission from ref. [61]. Copyright 2020 Elsevier.)

Zhang et al. [62] investigated the effect of different bias pressures on (TiVCrNb-SiTaBY)N HEA coatings prepared by magnetron sputtering. The results showed that the structure of the coating was a mixture of an amorphous phase and FCC crystal, and both unbiased and low-bias coatings had columnar crystal structures. With the increase in bias voltage, the structure gradually became uncharacteristic and the density increased, and this conversion also made the surface smoother. When bias voltage was applied, the hardness of the coating increased. As can be seen from Figure 13, under the condition of low bias voltage or no bias voltage, the wear of the coating was serious, and slightly abrasive wear occurred under a high bias voltage. Xu et al. [63] studied the effect of bias voltage on the superhard (AlCrTiVZr)N HEA coating synthesized by high-power pulsed magnetron sputtering. The results showed that the (AlCrTiVZr)N coating deposited at −150 V had a dense and unfeatured structure, the preferred orientation was (111), the minimum grain size was 11.3 nm, the residual compressive stress was −1.67 GPa, and the highest hardness was 48.3 GPa, reaching the superhard grade. Wang et al. [64] studied the effect of bias voltage on a CrNbTiMoZr HEA coating prepared by DC magnetron sputtering. It can be seen from Figure 14 that the microstructure of the coating changed from a columnar structure to an uncharacteristic structure with the increase in bias voltage, and the hardness of the coating was highest at −150 V, which was 9.7 GPa. However, the CrNbTiMoZr coating showed excellent tribological properties at low bias voltage.

Medina et al. [65] studied the effects of substrate temperature and bias on the CrMnFeCoNi HEA coating. The results showed that the coating had a mixture of HCP and BCC phases and intermetallic σ phase at higher substrate temperature. For the deposition process at room temperature, the diffusion of metal atoms was limited, and the hardness of the coating decreased with the increase in bias voltage. Dai et al. [66] studied the changes in the morphology and properties of FeCoCrNiMo0.3 HEA coatings prepared by magnetron sputtering by varying the substrate temperature and deposition time, and they found that the higher substrate temperature promoted coating densification and improved the pitting resistance of the coatings.

Dual-phase HEA has been proven to have the ability to overcome the strength–ductility balance. Li et al. [67] prepared dual-phase CuNiTiNbCr HEA coatings by high-power pulsed magnetron sputtering under different working pressures. It was found that the CuNiTiNbCr coating had a two-phase structure composed of an FCC matrix phase and Cu-rich BCC precipitated phase. The coating had a two-layer structure; a single FCC phase structure was near the substrate, and the upper part was an FCC + BCC phase structure. Compared with the FCC phase, the two-phase structure had a higher hardness. Kim et al. [68] studied the effects of different deposition parameters, such as sputtering power, deposition time, and reactive gas $N_2$, on the structure and mechanical properties of a TiZrHfNiCuCo HEA coating and nitride film. The results showed that a good nitride HEA coating could be obtained with a 300 W sputtering power, and the deposition rate was increased as well. XRD and TEM analyses showed that the structure of the HEA coating was amorphous, while the addition of $N_2$ active gas led to the formation of single-phase FCC nanocrystals in the HEA coating. TiZrHfNiCuCo HEA-nitrided film had a higher hardness and elastic modulus than the TiZrHfNiCuCo HEA coating. Khan et al. [69] studied the effect of deposition power on the microstructure and composition of the HEA coating, which showed that deposition power was an effective process parameter to control various properties of the AlCoCrCu0.5FeNi coating.

Through the reasonable selection of process parameters such as sputtering atmosphere, bias voltage, sputtering power, sputtering time, and substrate temperature, an HEA coating with excellent performance can be obtained, and each parameter is not independent of each other. Therefore, it is of great significance to explore the optimal parameter combination for the preparation of magnetron-sputtering HEA coatings.

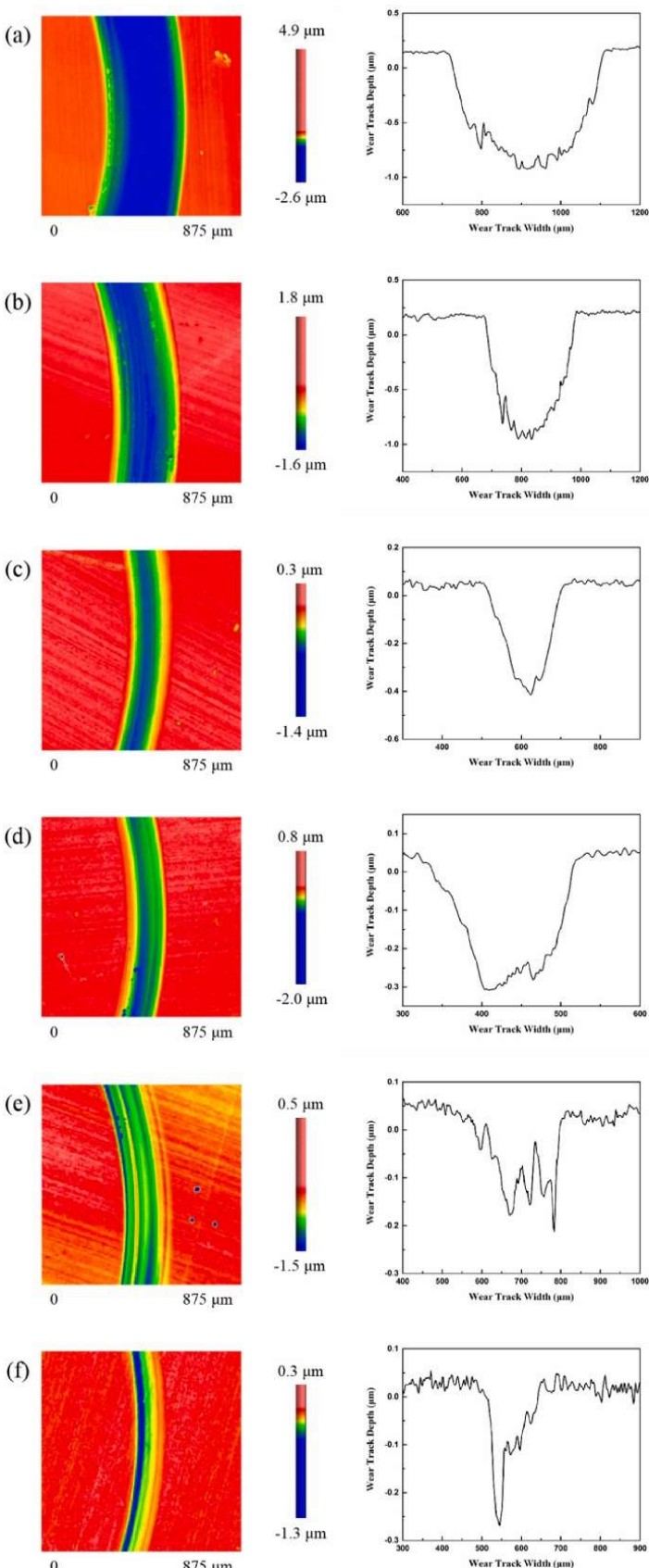

**Figure 13.** Morphology and 2D profiles of the wear tracks of the (TiVCrNbSiTaBY)N coatings deposited under bias voltages of: (**a**) 0 V, (**b**) −50 V, (**c**) −100 V, (**d**) −150 V, (**e**) −200 V, and (**f**) −250 V. (Reprinted with permission from ref. [62]. Copyright 2022 Elsevier.)

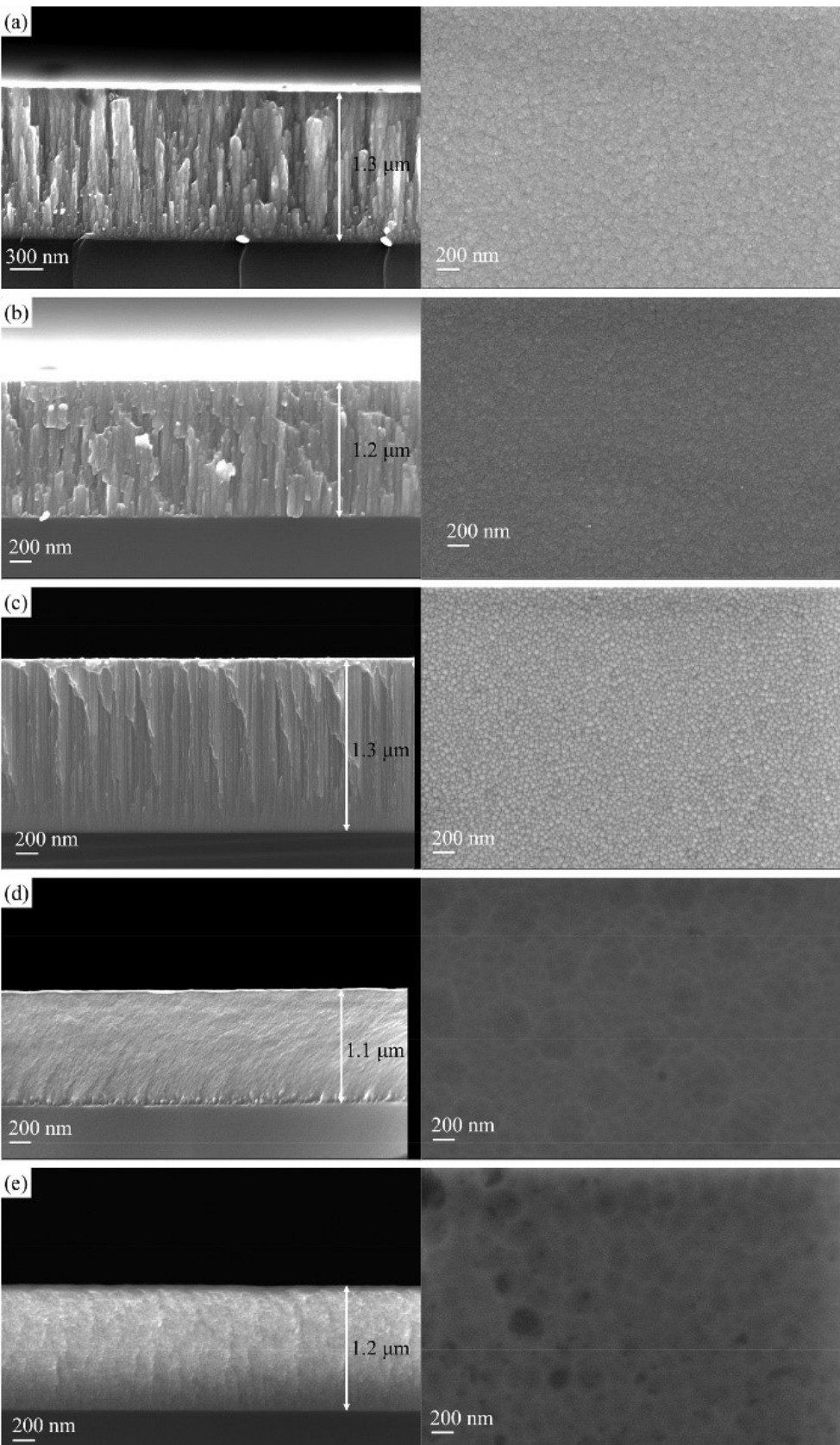

**Figure 14.** The SEM morphologies of cross-section and surface of CrNbTiMoZr films deposited at substrate bias of (**a**) 0 V, (**b**) −50 V, (**c**) −100 V, (**d**) −150 V, and (**e**) −200 V. (Reprinted with permission from ref. [64]. Copyright 2020 Elsevier.)

*5.3. Post-Processing*

At present, there are few studies on the microstructure transformation behavior of HEA and its coating after annealing and the effect of thermal shock on the coating in practical application, especially the effect of annealing temperature on the magnetron sputtering HEA coating. In fact, heat treatment can have a dramatic effect on the organization of the alloy. Wan et al. [70] studied the effect of annealing on the properties of the magnetron-sputtered AlCrNbSiTiV HEA nitriding coating. The results showed that the magnetron sputtering AlCrNbSiTiV HEA nitriding coating was mainly a face-centered cubic structure and had good high-temperature stability. When the annealing temperature was at 600 °C, the coating had high microhardness, low tool wear, and low surface roughness of the cutting workpiece. When annealed at 500 to 700 °C, the structure of the coating with good thermal stability and thermal shock resistance was relatively stable and the microhardness changed little.

## 6. Discussion

This paper summarizes three methods of preparing HEA coatings, such as thermal spraying, EDM deposition, and magnetron sputtering, and introduces the working mechanism and advantages and disadvantages of the three processes, focusing on combining the effects of the three processes, HEA coating composition regulation, process parameters, and post-treatment means, on the coating organization and properties.

This paper presents an overview of the preparation of HEA coatings by thermal spraying, electric spark deposition, and magnetron sputtering techniques and compares the advantages and disadvantages of the three processes. Thermal spraying technology has the advantages of extremely high energy density, low thermal distortion, low dilution rate of the matrix material, good interfacial bonding strength, and relatively low cost. However, the HEA coatings prepared by thermal spraying are prone to cracking due to high heat input. In addition, thermal spraying has certain requirements on the size and fluidity of the sprayed particles, and the particles are prone to oxidation due to high temperature. Electrospark deposition technology has the characteristics of fast heating and rapid cooling, small heat input, metallurgical bonding between coating and substrate, wide application range, and so on. The preparation of HEA coatings by this technology can give full play to the high entropy effect of multi-principal components, and can also form a solid solution phase with a simple body-centered cubic structure or face-centered cubic structure, and fine grains can achieve solid solution strengthening and fine grain strengthening, which is beneficial to obtain coatings with excellent properties. However, there are great differences in the electrical conductivity, thermal conductivity, and ductility of the materials, which have a great influence on the stability of the coating. Magnetron sputtering technology has the advantages of high speed, low substrate temperature, high bonding degree between coating and substrate, large-scale coating preparation, and so on. The coating prepared by this process has a high forming quality and uniform microstructure, but it requires strict working conditions and limits the thickness of the coating. Taking advantage of the advantages of different processes, combining two or more processes can lead to higher-quality HEA coatings. Electrochemical deposition is used as a synthesis method, which can control the phase composition and morphology of thin films with high precision [71,72]. With the increase in the thickness of the coating, the properties of the films were significantly improved. Electrochemical deposition does not require complex equipment and expensive raw materials, can be carried out at lower processing temperatures and lower-energy consumption, and can achieve low-cost preparation of HEA thin films on the substrate with complex geometry. The composition, morphology, and thickness of the films can be controlled by changing the deposition parameters. The preparation of HEA coatings by electrochemical deposition is a key development direction in the future.

It is well known that metal elements are easy to oxidize and corrode in high-temperature oxygen environments, and the oxidation corrosion of high-entropy alloy is even more serious in a high-temperature environment. On the one hand, it is related to the activity of the

metal elements themselves. On the other hand, the high-entropy alloy is in a multiphase corrosion environment; in addition to the oxidation gas and solid oxide corrosion in the environment, the multi-principal component composition makes the low-melting-point eutectic compounds form between different oxides, which makes the oxides easily liquefied or volatilized. Liquefied and volatile oxides can be used as a channel for rapid diffusion of metals and oxygen, strongly corroding high-entropy alloys. The structural and morphological properties of the alloy can be changed by doping with some oxidation-resistant elements to improve the high-temperature oxidation resistance [71,72]. Different element types and contents can have a great impact on the performance of HEA coatings. The addition of some elements with larger radii or self-fusing nonmetallic elements can effectively improve the alloy performance, and the doping of ceramic particles or rare earth elements can also improve the performance of HEA coatings to a certain extent. The effect of different element types and contents on the HEA coating varies; to achieve higher-quality HEA coatings, the mechanism of the influence of element type and content on the microstructure and properties of coatings should be further investigated.

The process parameters are very important to obtain high-quality coatings. The particle velocity, temperature, and particle size of thermal spraying have important effects on the dilution rate, density, and bond strength of the coating. When the particle has a high flight speed, it can form a dense coating. When the particle temperature is too low, the bonding strength between the coating and the matrix is insufficient. When the particle is semi-molten, the bonding strength between the coating and the matrix is improved. According to the discharge mechanism in the process of EDM deposition, the electric energy stored in the power supply is converted into heat energy in the form of a pulse. Parameters such as deposition voltage, deposition capacitance, and strengthening time directly affect the temperature of coating materials, and determine the melting state of HEA powder, grain growth time, and element diffusion rate. These have an important effect on the microstructure, bonding strength, and properties of the coating. In magnetron sputtering, sputtering atmosphere, bias, sputtering power, sputtering time, substrate temperature, and other process parameters have a direct relationship to the density and binding strength of the coating. Choosing appropriate process parameters is conducive to obtaining the HEA coating with excellent performance. Therefore, it is necessary to study the interaction between the parameters and the mechanism of the coating microstructure. The composite effect of various parameters is simulated by finite element analysis, which provides a theoretical reference for the preparation of more excellent HEA coatings.

The residual stress often causes the cracking of the coating and hurts the coating. During the hot forming process, residual stresses are mainly caused by the quenching of the deposited particles and the difference in the coefficient of thermal expansion between the deposited coating and the substrate. During the cold forming process, with the growth of the film, the particles arriving at the substrate will produce the mass transfer with the surface diffused. Under the influence of the substrate interface and grain interface, the atomic arrangement is incomplete and the formation of pores and other defects along with the grain nucleation growth increases the residual stress. Heat treatment of the HEA coating reduces the temperature gradient between the coating and the substrate, effectively eliminating residual stresses and reducing cracking. The preheating of the substrate reduces the temperature difference between the substrate and the coating, which is also very helpful in obtaining a defect-free coating. In addition, subsequent surface treatments, such as ultrasound-assisted mixing of elements in the coating, can avoid stress concentration and improve the quality of the coating.

## 7. Conclusions

Thermal spraying has the advantages of simple operation, wide preparation area, and on-site preparation, but requires high powder size and fluidity, and due to the high temperature, the preparation process requires a large amount of protective gas to prevent the particles from thermal oxidation. EDM and magnetron sputtering techniques produce

high-quality coatings with high bond strength and a more homogeneous coating structure, but they require harsher working conditions, are not easy to prepare on-site, and the thickness of the prepared coatings is somewhat limited. The microstructure and properties of HEA coatings prepared by different methods are bound to be different. Moreover, HEAs have a wide range of composition systems, and the differences in composition inevitably lead to differences in the microstructure and properties of the coating. The choice of process parameters and post-treatment methods will also affect the microstructure and properties of the coating. Further investigation into the regulation of HEA composition, parameter optimization, and post-treatment is key to obtaining high-quality coatings.

**Author Contributions:** Literature search, K.L., W.G. and J.Z.; study design, K.L., J.Z. and X.H.; data analysis, K.L.; writing, K.L.; data collection, J.Z. and W.G.; figures, X.H. All authors have read and agreed to the published version of the manuscript.

**Funding:** This work received the financial support of the National Natural Science Foundation of China (52004154), the Shandong Provincial Natural Science Foundation (ZR2020QE002, ZR2020ME047, ZR2020ME164), the National Key Laboratory Foundation of China (6142005190208), and the Shandong Provincial Key Laboratory of Precision Manufacturing and Non-traditional Machining.

**Institutional Review Board Statement:** Not applicable.

**Informed Consent Statement:** Not applicable.

**Data Availability Statement:** Not applicable.

**Conflicts of Interest:** The authors declare no conflict of interest.

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
