# Peer review of "Progress on New Preparation Methods, Microstructures, and Protective Properties of High-Entropy Alloy Coatings"

_coatings, doi:10.3390/coatings12101472_

Round 1

Reviewer 1 Report

Review report: Progress on new preparation methods, microstructures and protective properties of high entropy alloys coatings
1.      Shorten the length of the abstract section and add only key information.
2.      Discuss the Novelty and clear application of the work.
3.      Sentences are incomplete at few places. Check the manuscript thoroughly.
4.      Shorten the length of the introduction section and add key published work and try to make a bridge between current and previous published work. Refer some recently published work on coating: https://doi.org/10.1016/j.ceramint.2018.01.131; https://doi.org/10.1007/s41779-018-0258-4; https://doi.org/10.1007/s12540-020-00705-w
.

5.      Discuss the role of the residual stresses in coating to strengthen the quality of the manuscript.

6. Discuss about the author own contribution in area of the coating.

7. The interface layer presented in Fig. 7 needs more technical discussion. Also provide the major bond formation mechanism at the interface. Refer to following:

8. Process parameters sections need clear and depth discussion: DOI: 10.1520/JTE20180247

9. In conclusion section each conclusion consists of eth one paragraph which is not good for technical paper. Remove it and add only key bullet points.

Reviewer 2 Report

Referee Report

on paper “ Progress on new preparation methods, microstructures and protective properties of high entropy alloys coatings “ (coatings-1950600) by authors Kefeng Lu, Jian Zhu, Wenqing Ge, and Xidong Hui submitted to Coatings

This is interesting paper. It reviews the researches on the preparation methods of high entropy alloys coatings by thermal spraying, electrospark deposition technology, and magnetron sputtering techniques. There are few summaries about the preparation method, corresponding microstructures and protective properties of high entropy alloys coatings produced by thermal spraying, electrospark deposition technology and magnetron sputtering. This paper will introduce the working mechanism of each method, will compare the advantages and disadvantages of each method, will focus on the influences of the compositions, process parameters and post-treatment process on the microstructures and properties of the coating. Furthermore, this paper will outline the correlation between "preparation methods - process parameters - microstructure - properties, which will provide a reference for further development of the application of high-entropy alloy coatings. The presented data are reliable without any doubts. However, I have some questions. I would like to note a few points to improve the paper before it can be published:

1.    The authors should give in Introduction some information about other high entropy compounds:

(1). D.A. Vinnik, V.E. Zhivulin, E.A. Trofimov, S.A. Gudkova, A.Yu. Punda, A.N. Valiulina, M. Gavrilyak, O.V. Zaitseva, S.V. Taskaev, M.U. Khandaker, A. Alqahtani, D.A. Bradley, M.I. Sayyed, V.A. Turchenko, A.V. Trukhanov, S.V. Trukhanov, A-site cation size effect on structure and magnetic properties of Sm(Eu,Gd)Cr0.2Mn0.2Fe0.2Co0.2Ni0.2O3 high entropy solid solutions, Nanomaterials 12 (2022) 36. https://doi.org/10.3390/nano12010036.

(2). V.E. Zhivulin, D.P. Sherstyuk, O.V. Zaitseva, N.A. Cherkasova, D.A. Vinnik, S.V. Taskaev, E.A. Trofimov, S.V. Trukhanov, S.I. Latushka, D.I. Tishkevich, T.I. Zubar, A.V. Trukhanov, Creation and magnetic study of ferrites with magnetoplumbite structure multisubstituted by Al3+, Cr3+, Ga3+, and In3+ cations, Nanomaterials 12 (2022) 1306. https://doi.org/10.3390/nano12081306.

2.    Various methods for obtaining complex compound precursors are known:

(3). K.K. Kadyrzhanov, D.I. Shlimas, A.L. Kozlovskiy, M.V. Zdorovets, Research of the shielding effect and radiation resistance of composite CuBi2O4 films as well as their practical applications, J. Mater. Sci.: Mater. Electron. 31 (2020) 11729–11740. https://doi.org/10.1007/s10854-020-03724-w.

(4). T.I. Zubar, T.I. Usovich, D.I. Tishkevich, O.D. Kanafyev, V.A. Fedkin, A.N. Kotelnikova, M.I. Panasyuk, A.S. Kurochka, A.V. Nuriev, A.M. Idris, M.U. Khandaker, S. V. Trukhanov, V.M. Fedosyuk, A.V. Trukhanov, Features of galvanostatic electrodeposition of NiFe films with composition gradient: influence of substrate characteristics, Nanomaterials 12 (2022) 2926. https://doi.org/10.3390/nano12172926.

In the paper it is necessary to mention this information.

3.    For alloys the stoichiometry is particularly important. The deviation from stoichiometry and appearance of the oxygen anions can lead to a change in the charge state of the cations, which in turn will greatly change the electronical parameters. That will seriously affect the practical application of the materials obtained. What is the oxygen stoichiometry of prepared alloys samples? It is well known that the complex transition metal compounds easily allow the oxygen excess and/or deficit:

(5). S.V. Trukhanov, A.V. Trukhanov, A.N. Vasiliev, A.M. Balagurov, H. Szymczak, Magnetic state of the structural separated anion-deficient La0.70Sr0.30MnO2.85 manganite, J. Exp. Theor. Phys. 113 (2011) 819-825. https://doi.org/10.1134/S1063776111130127.

(6). A. Kozlovskiy, K. Egizbek, M.V. Zdorovets, M. Ibragimova, A. Shumskaya, A.A. Rogachev, Z.V. Ignatovich, K. Kadyrzhanov, Evaluation of the efficiency of detection and capture of manganese in aqueous solutions of FeCeOx nanocomposites doped with Nb2O5, Sensors 20 (2020) 4851. https://doi.org/10.3390/s20174851.

Effect of oxygen anions on the electronic properties of investigated samples should be discussed in 3. Results and discussion.

4.    The presented 6 papers should be inserted in References.

The paper should be sent to me for the second analysis after the moderate revisions.

Reviewer 3 Report

Dear authors, please see the remarks presented in the attached review document.

Round 2

Reviewer 1 Report

Accepted. 

Reviewer 3 Report

Dear authors, I see major improvements of your manuscript. You answered clearly and satisfactory to all my review remarks. I recommend publishing the manuscript in the MDPI journal.